

# LUCATOOv1 – A new land use change allocation tool and its application to the planetary boundary for land system change with the LPJmL model

Arne Tobian[1,2], Sarah Cornell[1], Ingo Fetzer[1,3], Dieter Gerten[2,4], Johan Rockström[2,1,5]

[1] Stockholm Resilience Centre, Stockholm University; Stockholm, Sweden
[2] Potsdam Institute for Climate Impact Research (PIK), Member of the Leibniz Association; Potsdam, Germany
[3] Bolin Centre for Climate Research, Stockholm University; Stockholm, Sweden
[4] Department of Geography, Humboldt-Universität zu Berlin; Berlin, Germany
[5] Institute for Environmental Science and Geography, University of Potsdam; Potsdam, Germany

**Correspondence to:** Arne Tobian (tobian@pik-potsdam.de)

**Abstract.** Anthropogenic alterations to terrestrial ecosystems resulting from land use transformation and agricultural intensification represent a significant driving force of global environmental change. The planetary boundary for land system change is one approach to determine an upper tolerable limit to such modifications, measured in terms of the remaining extent

of major forest biomes in temperate, tropical, and boreal climatic zones on the different continents. Here, we introduce a land use change reallocation tool (LUCATOO) that can accurately represent the spatial distribution of agricultural land use for different statuses and transgression levels of the planetary boundary for land system change. By representing such configurations of global land cover and land use patterns, the tool facilitates a systematic assessment of the impacts of afforestation and deforestation scenarios on the status of this and other interconnected planetary boundaries. LUCATOO has

been developed in the programming language R, is openly accessible, and can be readily adapted for land use change scenarios in applications beyond the planetary boundaries framework.

**Keywords:** Planetary boundaries, land use change, global environmental change, forest, deforestation, land system change

**Short summary**

The land use change reallocation tool LUCATOO enables the creation of future land use change scenario datasets tailored to specific requirements in model study applications. Its usability is demonstrated in the planetary boundaries interaction context. Being written in the programming language R and made openly accessible, LUCATOO can be easily adapted to be employed in contexts other than the planetary boundaries framework.



## 1 Introduction

Human activities exert a significant influence on a multitude of Earth system processes. The planetary boundaries framework delineates safe conditions for nine essential processes vital for maintaining long-term Earth system stability (Richardson et al., 2023; Rockström et al., 2009; Steffen et al., 2015). For each planetary boundary (PB), one or more control variables indicate the level of boundary approach or transgression as caused primarily by human impact. The individual PBs are situated at the lower end of the scientific uncertainty range concerning the consequences of human interference with each of the critical

Earth system processes, in accordance with the precautionary principle (Kriebel et al., 2001). They delineate a safe zone from a zone of increasing risk followed by a zone of high risk (Steffen et al., 2015). The most recent comprehensive assessment indicates that human pressures already exceed the PBs for six of the nine critical processes (Richardson et al., 2023).

The transformation of terrestrial ecosystems and subsequent loss of major forest biomes through land use and land cover change (LULCC) primarily for agricultural expansion has been identified as one of the PBs. This planetary boundary for land

system change (PB-LSC) - defined by the remaining forest extent of the principal biomes, tropical, temperate and boreal compared to potential natural vegetation (PNV) both on a regional and global level - is found to be strongly transgressed (Richardson et al., 2023). Forests play an important role in maintaining Earth system processes. They provide space for hosting biodiversity (Brockerhoff et al., 2017), regulate the water cycle (Pranindita et al., 2022), and capture and sequester carbon (Sedjo and Sohngen, 2012). Fundamentally, they are essential for mitigating climate change (Snyder et al., 2004; West et al.,

2011). The alteration of forest from LULCC has been identified as a primary anthropogenic pressure on ecosystems worldwide (Ostberg et al., 2015) resulting in significant perturbation to global energy and moisture fluxes as well as nutrient cycling (Verburg et al., 2015). LULCC, especially when it results in forest loss, imposes pressures on a number of essential Earth system processes (Fig 1, supplement section S2). Consequently, historical LULCC has been significantly contributing to breaching several PBs (Campbell et al., 2017; Gerten et al., 2020; Rockström et al., 2020).

Yet, the extent to which forest cover must remain intact to sustain safe planetary conditions remains largely unclear. Currently, the boundary value is based on the assumption that 50% of PNV for temperate forests, and 85% of tropical and boreal forest respectively are to remain intact in all regions of their occurrence (Richardson et al., 2023; Steffen et al., 2015). However,





extensive deforestation in most parts of the world over the last centuries has resulted in a significant transgression of PB-LSC on a global, continental and biome scale (Richardson et al., 2023).

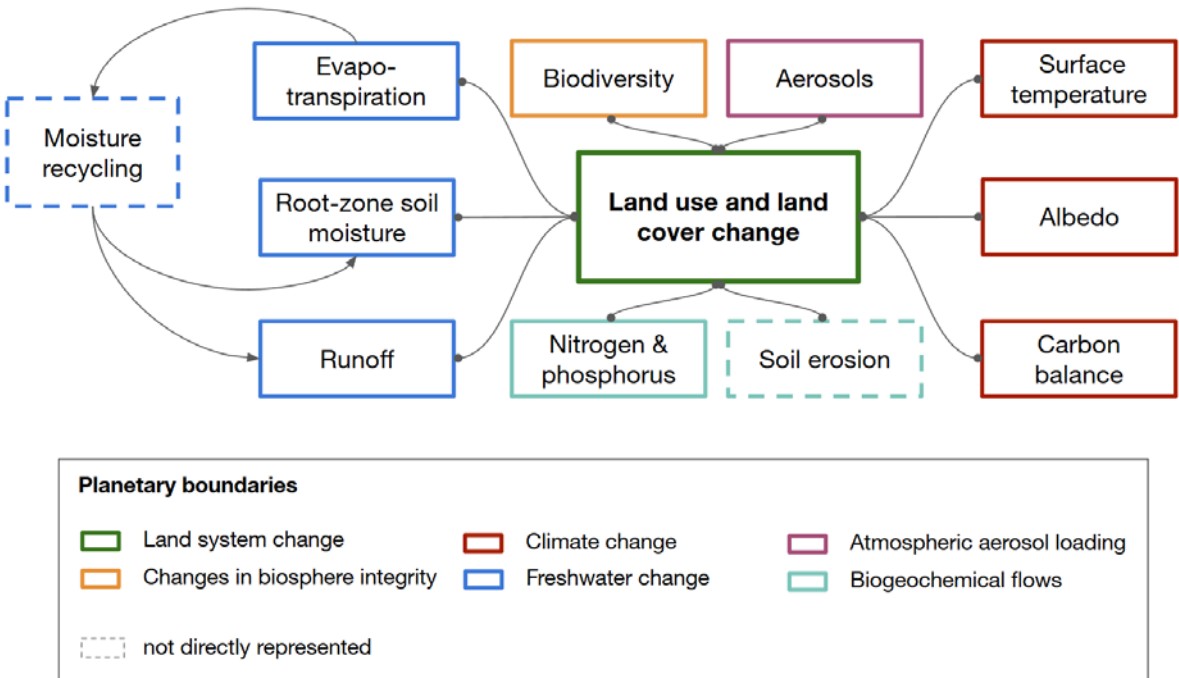


**Figure 1: Schematic overview of how LULCC is at the nexus of various critical Earth system processes, with PB-LSC being coupled to other PBs. Note that not all interactions are shown. Soil erosion is included here but not represented by a specific PB (but implied in the phosphorus sub–boundary of biogeochemical flows). Moisture recycling is indirectly included in the freshwater change PB. The arrows connecting evapotranspiration to moisture recycling and root–zone soil moisture and runoff show the direction of**
**dependency.**

In this context, an analysis of the consequences of future LULCC on the stability of the Earth system, as understood through the multiple aspects encompassed in the PB framework, is of utmost importance. Specifically, it is unclear how different levels of human impacts on the land system would affect the status of other PBs. The existing projections regarding future LULCC
are characterized by a considerable degree of uncertainty due to the wide array of assumptions employed in the allocation models utilized (Alexander et al., 2017). It is noteworthy that the areas exhibiting a heightened level of uncertainty in both the magnitude and allocation of projected land use are primarily located along the borders of globally significant biomes, such as boreal and tropical forests (Prestele et al., 2016). Consequently, these regions have the potential to significantly influence PB-LSC.

This underscores the necessity of an examination of future modifications to the land system across a range of spatial and temporal scales and biomes. However, as future scenarios of LULCC often rely on e.g. the Shared Socioeconomic Pathways,





they do not directly align with the PB-LSC definition (Bukovsky et al., 2021) and can thus not be employed to study PB-LSC and its interactions with other PBs. In consequence, the development of a methodology for explicitly obtaining spatial scenarios of LULCC that are tailored to the definition of PB-LSC is required (this enables the identification of areas where intense PB

interactions occur, thereby providing valuable insights for stakeholders in the implementation of the framework). In this study, we introduce the 'land use change reallocation tool' LUCATOO, an algorithm that enables the specific rearrangement of LULCC patterns in a spatially explicit manner following environmental and scenario-driven constraints. This is done to match the definition of PB-LSC and potential scenarios of its transgression globally, on the different continents, and for the different forest types. Although LUCATOO has the potential to be used for generating LULCC patterns for other scientific inquiries,

its primary application demonstrated in this work is to simulate various PB-LSC statuses within the three major forest biomes: tropical, temperate, and boreal on a regional level. The tool has been constructed to operate with data pertaining to the distribution of natural and agricultural vegetation, as output from a Dynamic Global Vegetation Model (DGVM), here LPJmL version 5 (Schaphoff et al., 2018a). The generated LULCC datasets can then be employed for instance to prompt DGVMs or Earth System Models to quantify the impacts of the specific PB statuses, including the resulting changes in the interlinked

PBs, with the view to assessing the impacts on Earth system stability, as demonstrated by Richardson et al. (2023) who applied land use data generated by LUCATOO in this context.

## 2 Methods

### 2.1 Development of land system change scenarios

The following develops three land system change scenarios that depict the various statuses for PB-LSC which are to be matched by LULCC datasets provided by LUCATOO. These scenarios represent crucial inflection points for the control variable status (see details Fig 2), as outlined by the planetary boundaries definition (Richardson et al., 2023; Steffen et al., 2015).

I.    *At the planetary boundary:* limiting LULCC to match PB-LSC in order to maintain long-term stability and Holocene-like Earth system conditions. This scenario entails the preservation of 50% of the potential area covered by temperate

biomes and 85% of both the tropical and boreal forest biomes, respectively.

II.    *Increasing risk:* the scenario which represents the upper end of the zone of increasing risk (yellow zone, Fig 2). This occurs when breaching the boundary and results in a loss of Holocene-like Earth system characteristics. It is set at a remainder of 30% for the temperate and 60% for tropical and boreal forest.


III.    *Strong transgression:* in this scenario, where LULCC is deeply in the high risk zone, it is assumed that only 20% of the temperate and 40% of the tropical and boreal forest remain.



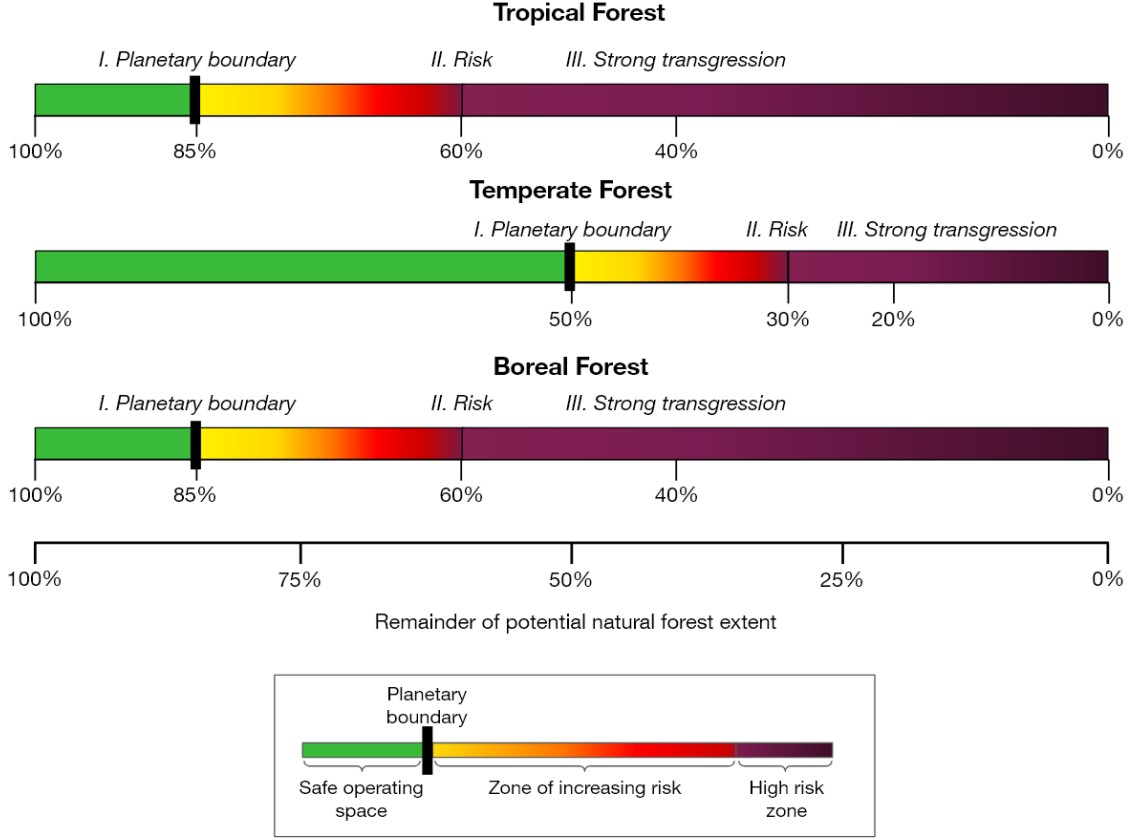

**Figure 2: Scheme of three different PB-LSC scenarios associated with different levels of human pressure on PB-LSC as defined by Steffen _et al_. (2015) and Richardson _et al_. (2023). The horizontal axis indicates the remaining forest extent of the three major forest biomes (tropical, temperate and boreal). The colors of the bars show how much forest of each biome on each continent must remain in order to stay within 'safe' (green), 'increasing risk' (yellow) and 'high risk' (red) state. In the order of increasing human pressure, the three derived LULCC scenarios are (i) at PB-LSC, (ii) at the transition zone between increasing and high risk and (iii) at a position of dangerous risks of a strong transgression level.**

-------------------------------------------------------------------------------------------------------------------------

**Box 1 - LPJmL model description**

-------------------------------------------------------------------------------------------------------------------------

LPJmL (Lund-Potsdam-Jena with managed land) is a state-of-the-art DGVM that operates cell-based at a spatial resolution of 0.5° x 0.5° and runs on daily time steps (von Bloh et al., 2018; Schaphoff et al., 2018a, b). It encompasses comprehensive and interlinked hydrological, carbon and nitrogen cycles. Natural vegetation is represented by 11 individually parameterized plant functional types (PFTs). Eight of those are woody species from the tropical, temperate and boreal biomes who differ by their morphological and phenological strategies, fire and heat resistance and bioclimatic optima. The establishment, fractional coverage, productivity and survival of local tree stands is the product of environmental factors (e.g. climate and availability of



water) and natural vegetation dynamics, such as heat mortality, competition, access to resources (Schaphoff et al., 2018a).

LPJmL features a managed land model where LULCC is prescribed by a land use dataset. Here, we use historical land use patterns (up until the year 2015) by Frieler *et al.* (2017), based on HYDE 3.2 (Goldewijk et al., 2017). LULCC is represented by 16 different crop functional types (CFTs), involving cropland, pasture/managed grass, bioenergy plantations as well as other types of land use (see supplement section S1 for a full list). Besides purely rainfed cultivation, each CFT can either be surface-irrigated, sprinkler-irrigated or drip-irrigated to compensate for insufficient rainfed water supply, thereby providing a

total of 64 different CFT-irrigation combinations. The cropland extent and cell-specific CFT distribution is prescribed on an annual basis. LPJmL is extensively validated (Schaphoff et al., 2018b) and has previously proven to be suitable for studying the impacts of LULCC in the PB context (Gerten et al., 2020; Heck et al., 2016, 2018a; Tobian et al., 2024a).

For the application herein, LPJmL is forced with historical climate data provided by the GFDL-ESM4 physical climate model (Held et al., 2019) from the latest climate model intercomparison project CMIP6 (Eyring et al., 2016) and has been bias-

corrected (Lange, 2019). Nitrogen deposition scenarios were taken from the ISIMIP3 database (Jägermeyr et al., 2021).

---------------------------------------------------------------------------------------------------------------------------------

## 2.2 Simulations

The mathematical algorithm of the LUCATOO tool builds on two simulations that quantify the extent of the three principal

forest biomes on each continent under two scenarios: (i) PNV, representing the reference status of the maximum potential forest extent and (ii) current LULCC, used to assess the remaining forest cover at the end of the historical period. An overview is provided in Fig 3.

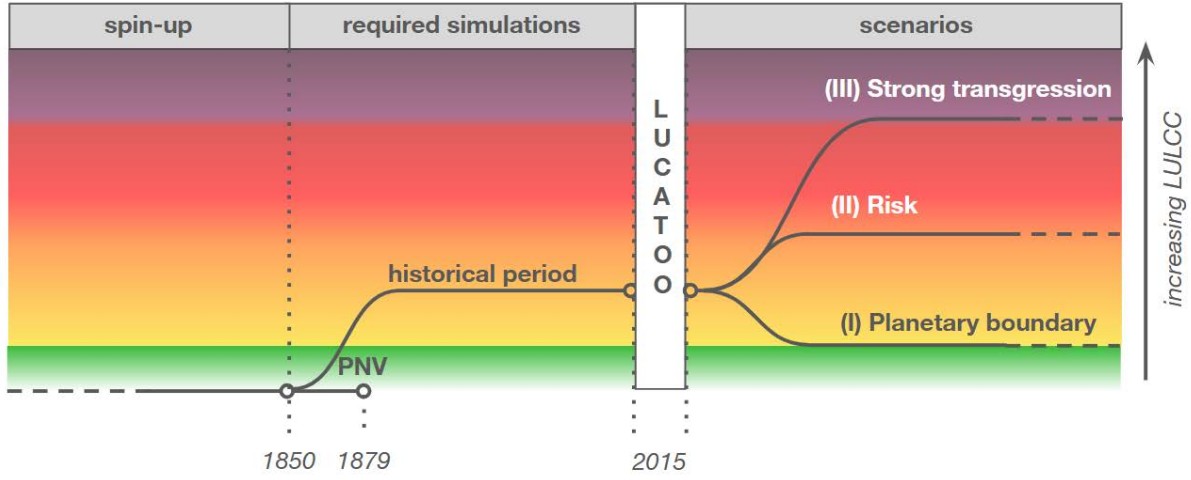





**Figure 3: Overview of the underlying LPJmL simulations and subsequent scenarios, their LULCC values and corresponding statuses of PB-LSC (following the traffic light color scheme introduced in Fig 2). A spin-up period precedes the potential natural vegetation (PNV) and historical climate change runs. The LUCATOO scenarios (on the right) are the product of LULCC reallocation, starting from 2015. The historical and scenario LULCC extent is smoothed out for illustrative purposes.**

Prior to the two simulations, the LPJmL model undergoes an 8000 year spin-up period of repeated pre-industrial climate (1601

- 1849) and without LULCC. During this spin-up period, carbon and water pools as well as natural vegetation (PFTs) are

established at a bioclimatic equilibrium (cf. Box 1 - LPJmL model description). A subsequent spin-up is required for the

historical LULCC simulation. Here, 390 years of repeated pre-industrial climate for 1849 introduces agriculture and historical

LULCC, leading to the cell-based distribution of CFT and irrigation schemes and the adjustment of carbon and water pools

under LULCC (historical land use data from 1700 - 2015 are obtained from Frieler et al. (2017)). LULCC is expressed by the

occurrence and relative proportion of crop functional types (CFTs) and their respective management options in each gridcell

('CFT-mix', based on the LPJ grid).

The PNV simulation is forced by historical climate (1850-2015) but excludes any form of LULCC. The spatial distribution of

PNV biomes is determined as the mean state over the first 30 years of this simulation period (1850 - 1879). In contrast, the

historical LULCC simulation is forced by historical climate and land use over the same time span (1850-2015). The remaining

forest biome extent at the end of this simulation (year 2015) serves as the basis for assessing the current status of PB-LSC, as

elaborated in the subsequent section.

To identify the PNV biome distribution and its current remainder, the PFT distribution in all grid cells need to be processed.

The biome classification scheme applied here is a slightly modified version of the framework developed by Ostberg et al.

(2013), as adapted in Tobian et al. (2024a). Subsequently, the current status of the regional boundary for PB-LSC is evaluated

by quantifying the extent of remaining PNV forest across each biome on each continent. This evaluation sets the starting point

for LUCATOO.

## 2.3 Simulations

The distribution of natural vegetation (PFTs) within a cell is determined by vegetation dynamics simulated by LPJmL (see

Box 1). Consequently, only anthropogenic LULCC (represented by CFTs) can be modified and reallocated to influence the

PB-LSC status.

Importantly, the global PB-LSC scenario targets must be achieved at regional scales. For instance, if a scenario stipulates that

80% of PNV coverage is required for tropical forests, this threshold must be met independently on each continent where the

biome is present.

In alignment with the PB-LSC scenario (see Fig 2), the LUCATOO algorithm adjusts LULCC by either i) facilitating

afforestation by reducing the area allocated to CFTs within the biome PNV extent, or ii) implementing deforestation by

increasing CFT coverage at the expense of forest PFTs. Deforestation can result from intensifying LULCC in cells where CFTs

are already present, or by expanding LULCC into currently pristine cells (where CFTs are absent). In accordance with the PB-



LSC definition, LULCC allocation occurs strictly within the spatial extent of the biome under PNV. The algorithm iteratively
processes each scenario, adjusting the CFTs extent according to the specified parameters. For each scenario, the extent of each
biome is adjusted individually on each continent to ensure alignment with the regional PB-LSC definition (Richardson et al.,
2023).

Finally, the algorithm produces a LULCC dataset in which CFTs have been reallocated to reflect the requirements of the
respective PB-LSC scenario. The following section outlines the methodology of LULCC reallocation employed by the
algorithm in conjunction with its application in LPJmL.





**Figure 4. Cell-based depiction of a) the reduction process for afforestation and b) the intensification and c) the expansion operations for deforestation. Green blocks illustrate the occurrence of PFTs (natural vegetation as simulated by LPJmL), orange blocks the presence of CFTs (prescribed anthropogenic land use). While the reduction operation yields a contraction of LULCC in a gridcell (favoring a gain in PFT area), the intensification operation replaces natural vegetation by CFTs.**

In scenarios requiring a reduction in LULCC - such as returning to the PB-LSC for biomes and regions where it is currently transgressed - afforestation (Fig 5, pathway 1) is applied. In this case, all cells within the regional PNV biome extent are uniformly adjusted using a reduction factor, decreasing the cell-specific CFT-mix to achieve the required biome extent target





on each continent (see Fig 4a). A flag has been implemented in LUCATOO to prohibit deforestation in areas where the PB-LSC status is currently below the target value (i.e., where the remaining PNV forest exceeds the defined PB-LSC threshold). Conversely, deforestation (Fig 5, pathway 2) is employed when the objective is to intensify the LULCC pressure, representing a further transgression of PB-LSC. This pathway consists of two iterative steps: intensification and expansion.

First, the intensification process (Fig 4b) focuses on cells where anthropogenic land use is already present but forest biome PFTs still exist. Here, LULCC intensification replaces PFTs with CFTs, exerting increasing pressure on PB-LSC. The existing CFT-mix is multiplied by an intensification factor, applied uniformly across all cells within the PNV biome extent, analogous to the reduction factor used in the afforestation pathway. In some cells, this may result in CFT coverage exceeding the available area (i.e., surpassing 100% of the cell's capacity). Any excess LULCC is subsequently reduced and carried forward to the next

step of the iterative deforestation process (cf. Fig 4b).

Second, the expansion process (Fig 4c) permits the spread of CFTs into cells currently unaffected by LULCC. Each pristine cell (i.e., with no CFTs) within the targeted biome is assigned a CFT-mix based on the CFT composition of its eight neighboring cells, where applicable (as illustrated in Fig 4c). This step uses a smoothing function that extends each CFT type beyond the cell's periphery to the adjacent cells. Prior to expanding CFTs into a pristine cell, the bioclimatic suitability of the

specific CFT is evaluated, and unsuitable areas such as wetlands and waterways are excluded (as these can not be cultivated in LPJmL). The algorithm also allows cells to be designated as unsuitable for LULCC expansion, for example, to preserve intact and connected forest biomes. In subsequent iterations, newly cultivated cells can undergo further intensification and LULCC is further expanded until the land use requirements are met.

LUCATOO iteratively loops over all biomes across all continents until the scenario-specific requirements are fulfilled and the

LULCC dataset can be exported. As a final step, the difference between the original LULCC dataset and the adjusted, scenario-specific dataset is linearly interpolated over a predefined period to enable a transient simulation. The resulting datasets can then be employed as input for models such as LPJmL or other applications aimed at assessing the impact of different scenarios on the state of other PBs.



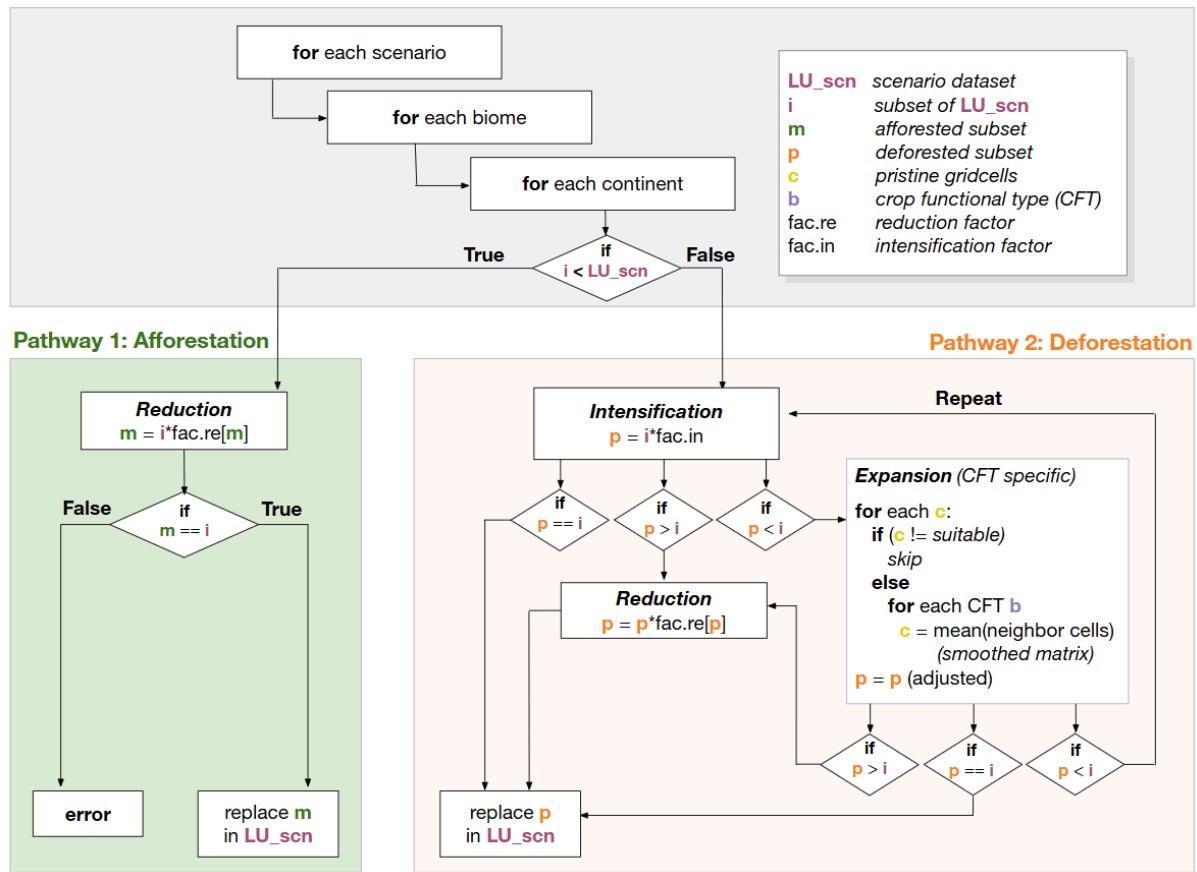

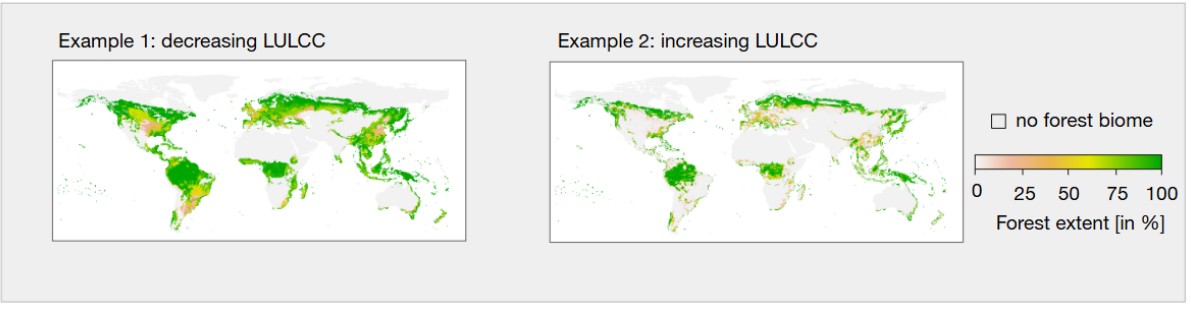


**Figure 5. Depiction of the functionality of LUCATOO. The upper gray box is defined by the scenarios of its application. The algorithm loops over each PB-LSC scenario (see methods) to adjust CFTs accordingly. Land use is modified on each continent and for each forest biome (tropical, temperate and boreal). If it is higher than the scenario-required LULCC, the current LULCC is reduced within the PNV extent of the specific biome and continent to provide space for the forest to extend into (pathway 1). The**
**opposite is the case if the current LULCC is smaller than the scenario requirements. Here, LULCC is expanded via deforestation through two iterative steps, intensification and expansion until the requirements are met (pathway 2). A detailed description of the pathways and their individual steps can be found in the method section of this paper and in the annotations of the code. After looping over each biome on each continent to adjust LULCC, the final LULCC dataset is exported and can act as an input for application studies.**



## 3 Results

Under current land use (2015, end of the historical period simulation) most areas have already crossed the biome-specific boundary values (see predominantly yellow and red colors in Table 1) which roughly matches earlier assessments (Gerten et al., 2020; Heck et al., 2018a; Richardson et al., 2023; Steffen et al., 2015; Tobian et al., 2024a). Differences emerge from the biome classification protocol, employed LULCC data set, chosen climate forcing, model version and PNV calculation (refer
to supplement section S4).

**Table 1. Remaining extent (in %) and status (in color) of the major forest biomes on each continent under LULCC in 2015. The numbers are colored based on the PB-LSC status as shown in Fig 2. Note: Americas: North and South America; Oceania: Australia and New Zealand (delineated after Heck et al. (2018a)).**

|  | Europe | Asia | Africa | Americas | Oceania |
|---|---|---|---|---|---|
| **Tropical** | NA | 73% | 64% | 73% | 79% |
| **Temperate** | 45% | 61% | 33% | 47% | 59% |
| **Boreal** | 67% | 62% | NA | 71% | 55% |


Returning to the *(I) Planetary boundary* for land systems requires large-scale afforestation in most regions to reestablish 50% of temperate and 85% or tropical and boreal PNV forest extent. There is a particularly strong need for afforestation in the European and Asian boreal forest biome and African tropics, if the PB ought to be maintained there. This is reflected by a strong reduction in CFT coverage, as shown in the central and right columns of Fig 6. Note that the prohibiting deforestation
flag mentioned in the methodology sections preserves the current forest remainder if the extent is larger than required by the scenario.

The *(II) Risk* scenario (situated at the inflection point between the increasing and high risk zone of PB-LSC) encompasses a remainder of 30% temperate and 60% boreal and tropical forest. In our LPJmL-based assessment, only the boreal forest in Oceania (red color in table 1) is within the high risk zone for LULCC in 2015, but note that boreal forest is only present in a
small area in New Zealand. We also found that most areas are within 'increasing risk' status (Table 1). Reallocating LULCC to meet the scenario leads to deforestation and expansion of agricultural coverage area (cf *(II) Risk* row in Fig 6). The Asian temperate forest illustrates that this effect is particularly strong in areas that have not yet breached the regional PB.

The *(III) Strong transgression* scenario would imply large-scale deforestation due to corresponding expansion of LULCC, causing a particularly strong reduction of forest extent around the currently intact forest biomes. This can be seen by the
simulated poleward deforestation of the boreal forest biome and loss of potential tropical forest at the edges of the Amazonas and African rainforest in the Congo Basin.



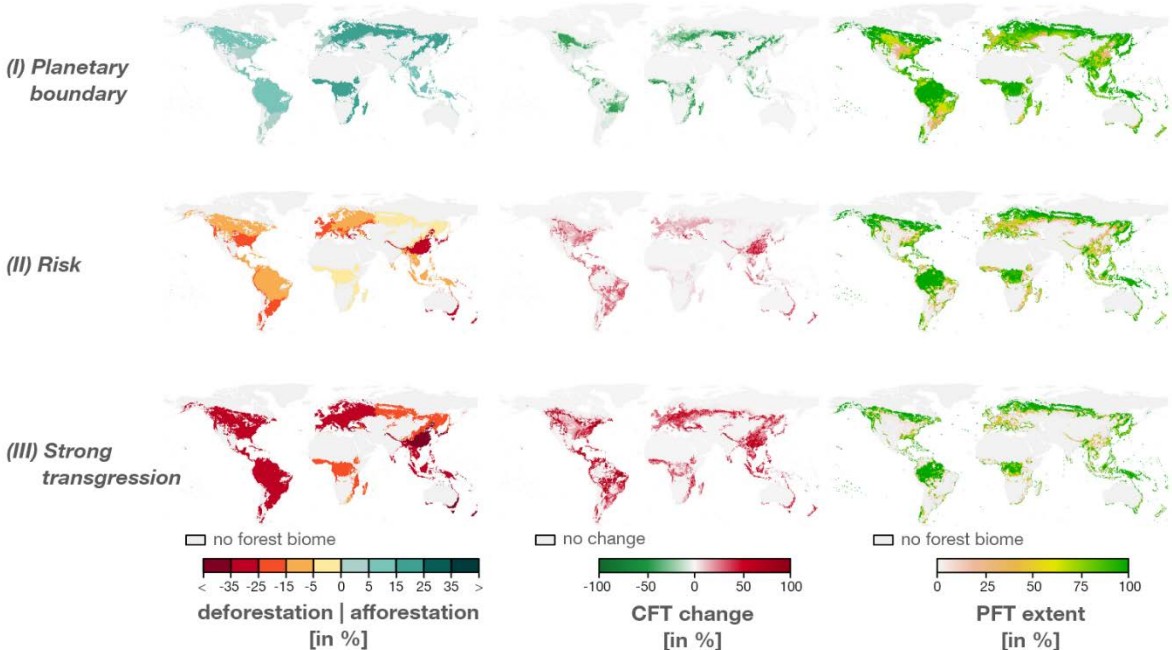

**Figure 6. Global LULCC maps derived from LUCATOO for different scenarios of PB-LSC maintenance and transgression:** *(I)*
*Planetary boundary*, *(II) Risk* **and** *(III) Strong transgression*. **Left panel: degree of required deforestation (red colors) or afforestation**
**(blue colors) per biome on each continent to meet the scenario. Central panel: resulting cell-based changes in crop functional type**
**(CFT) coverage in absolute percentage. Right panel: expected biome extent (calculated as the PNV extent with the CFT land cover**
**subtracted).**

## 4 Discussion

Being easily extendable and reproducible, LUCATOO is a versatile tool designed to bridge the conceptual gap of an adjustable
depiction of specific anthropogenic pressure levels in LULCC modeling. As such, it ensures the consistent and spatially explicit
mapping of different PB-LSC statuses while maintaining the flexibility to be employed for various applications outside the PB
framework context.

### 4.1. Applications within the planetary boundaries framework

The first scenario - returning to PB-LSC - can help to corroborate the present assumption that the 50% and 85% thresholds,
and inflection points to high-risk zones are well placed in terms of impacts of its transgression, e.g. in that it should not cause
a further transgression of other PBs. Applying LUCATOO in their model analysis (however lacking a detailed sensitivity test
of the thresholds), Richardson et al. (2023) already suggested that maintenance of PB-LSC is one prerequisite to avoid further
transgression of the climate change PB, while another study conversely suggested that the latter needs to be maintained to
avoid further transgression of the former (Tobian et al., 2024a).



The increasing risk and especially the high risk zones are characterized by a high degree of scientific uncertainty. Mapping out the zone of uncertainty with dedicated LULCC scenarios is crucial to improve both the robustness and precision of the safe boundary quantification. Especially since the studies that were used to set PB-LSC are scarce, potentially no longer state of the art and were never conducted for identifying a planetary boundary for land system change in the first place (Snyder et al., 2004; West et al., 2011). Further, the differences resulting from employing varying LULCC datasets and DGVMs need to be systematically addressed. Besides forests, other biomes which are currently missing in the PB-LSC definition - such as grasslands or savannas - could further be replaced by LULCC to study the impact of their transformation on the Earth system. In light of these current uncertainties behind the definition of PB-LSC and its thresholds, we see a strong potential for our tool to facilitate future work for an improved quantification and potentially extended definition of PB-LSC that goes beyond the current forest-based approach.

As highlighted in Fig 1 (and elaborated in supplement section S2), PB-LSC is tightly linked to other PBs. While most linkages have not yet been studied in the context of the planetary boundaries framework, LUCATOO has already found an application in a few studies. In Richardson et al. *(2023)* it was used to derive land use scenario datasets that build the foundation for studying interactions among the PBs for land system change and climate change in the POEM Earth model (Drüke et al., 2021). In particular, the study shows that the trajectory of the terrestrial carbon stock and land surface temperature are highly affected by the PB-LSC status, thereby highlighting that PBs have to be considered together. A more detailed analysis of the long-term impact of the LUCATOO-derived land system change scenarios on the Earth system was recently conducted by Drüke et al. (2024).

Moreover, the identification of hotspots of planetary boundary interactions is a crucial step to understand systemic risks, evaluate corporate environmental impacts and guide policy makers (Crona et al., 2023; Lade et al., 2021). LUCATOO opens a new realm of research to study the impacts of LULCC in the context of planetary boundary interactions and systemic environmental impacts across various sectors of human activity.

### 4.2. Potential extensions and applications

Our land use change reallocation tool is easily extendable to fit other applications, including consideration of different spatial extents (e.g. national or continental level), or studying the impacts of deforestation and afforestation alike to provide more in-depth analysis of links between LULCC and other Earth system processes as highlighted above (Fig 1).

Within the deforestation loop, the 'blocked cells' flag under the suitability check allows for protecting a selection of pristine cells (i.e. where no CFTs are present) from being deforested (Fig 4). For example, cells that are part of intact forest biomes and thus hotspots of forest biodiversity (Potapov et al., 2008) could be preserved as has been done in an earlier land use optimization study for staying within PB-LSC (Heck et al., 2018b). LULCC is a key disturber of terrestrial moisture recycling as it compromises the evapotranspiration flux that maintains it (te Wierik et al., 2021). Another application of the 'blocked cells' flag could thus be to study the impact of LULCC on moisture recycling in selected areas, for example by protecting or,



conversely, selectively deforesting the *most influential precipitationsheds* (moisture recycling source hotspots) building on
Weng et al. (2018).

The spatial extent of LULCC adjustment can also be varied. Following the definition of PB-LSC (Richardson et al., 2023; Steffen et al., 2015) we here adjust LULCC per forest biome on each continent. But the tool can also be adjusted to operate on a IPCC reference region level (Iturbide et al., 2020) as has been successfully tested (see supplement section S3 for details).

The expansion unit (Fig 4c) spreads agriculture to pristine cells and puts them under LULCC. As such, it only expands agriculture to cells neighboring existing agriculture and spares cells that are at a further distance from agriculture (explaining the strong reduction of forest extent around the edges of currently intact forest biomes as can be seen in Fig 6, central column). By placing the expansion unit behind the intensification, meeting the deforestation requirements is predominately achieved through intensifying LULCC in cells where agriculture is already present. In light of the ongoing land-sharing vs. land-sparing debate (Fischer et al., 2014) the approach taken here is inclined to follow the land-sparing strategy. This strategy favors intensification and minimizes the extent of farmland to set aside land for conservation purposes (Green et al., 2005), which is further endorsed by the option to block cells, e.g., from intact forest biomes. The order of the operations (intensification and expansion) could be reversed and a threshold for maintaining a certain percentage of PFTs per cell could be easily encoded into the algorithm to align it with the land sharing strategy which maintains a degree of biodiversity in less agriculturally intense cells. The resulting changes in carbon pools, yields and water fluxes could be compared to enrich the land-sharing vs land-sparing debate.

Lastly, both the status calculation and the placement of PB-LSC are affected by some (LPJmL-specific) assumptions and data choices (see supplement section S4 for a detailed remark).

## 5. Conclusion

LULCC is a major driver of environmental change in the Anthropocene and affects numerous key Earth system processes. Many of these processes are captured by the planetary boundaries framework, which sets a safe operating space for humanity to maintain planetary stability.

Currently existing LULCC scenario products are not aligned with the definition of PB-LSC and cannot be adjusted or modified to depict specific anthropogenic pressure levels. LUCATOO fills this gap and provides a foundation for assessing the strength and direction of PB interactions that result from varying pressure levels on PB-LSC. We invite future research to employ the tool to selectively afforest and deforest biomes to obtain a more robust boundary quantification that can extend the current forest-based definition to other biomes. Moreover, the algorithm can be tailored to fit LULCC-based research questions outside the context of the planetary boundaries framework.



*Code and data availability.* The open-source code of LUCATOOv1 is openly available at https://doi.org/10.5281/zenodo.14525229 (Tobian et al., 2024b). The LPJmL5 data that support the findings of this study are available upon reasonable request from the authors.

*Author contributions.*
Developing project ideas: AT, DG, IF, JR, SC
Modelling: AT
Paper writing: AT, DG, IF, SC, JR

*Competing interests.* The authors declare that they have no conflict of interest.

*Acknowledgements.* We thank our colleague Fabian Stenzel for reviewing the code and the team behind the LPJmL model for its continued efforts and hard work. Code annotation has been facilitated by using the AI tool Github Copilot.

*Financial support.* Arne Tobian received funding from the European Research Council through the 'Earth Resilience in the Anthropocene' project (No. ERC-2016-ADG 743080).

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
