# Peer review of "LUCATOOv1 – A new land use change allocation tool and its application to the planetary boundary for land system change with the LPJmL model"

_EGUsphere, 2025_

## Author Comment (AC1)

**RC1: 'Comment on egusphere-2025-2202', Anonymous Referee #1, 12 Aug 2025**

Replies to **Reviewer 1:**

- **General comment 1: This study introduces a new land allocation tool to produce spatially explicit datasets consistent with different states of the LSC planetary boundary. The conceptual framework is enticingly simple and forms a novel contribution. Using the PB framework, the authors construct unique scenarios which provide a much-needed contribution to the currently limited diversity of LULCC scenarios in the literature. However, the simplicity of the approach is not supported with sufficient discussion about its potential limitations. In particular, the manuscript would benefit from further discussion about the land use allocation algorithm employed which omits key factors known to influence LULCC, including land productivity, land use and conversion costs, and socioeconomic factors. Including these factors could result in significantly different patterns of LULCC than those simulated here and add further layers of uncertainty not explored.**

  **Response:** We thank the reviewer for their generally positive review and agree that the discussion is currently lacking a limitations section. In the revised manuscript, we will include a new limitations subsection that explicitly synthesizes the main points raised by the reviewer. In line with the reviewer remarks, we will stress that socioeconomic factors driving LULCC are vast, ranging from existing transport infrastructure that limits accessibility, to policy and legislation frameworks, as well as national and international demand for produced commodities. LUCATOO deliberately abstracts from these complex socio-economic drivers by applying a simplified and spatially uniform LULCC allocation scheme. This abstraction is intentional and motivated by the objective of generating internally consistent, spatially explicit land-use configurations that meet predefined planetary boundary–related constraints, thereby enabling controlled experiments on biophysical Earth system responses. These limitations will be explicitly discussed, and we will emphasize that the resulting scenarios should be interpreted as stylized, boundary-consistent land-system states rather than plausible future trajectories.

Representing socio-economic drivers and producing realistic future LULCC projections lies beyond the scope of the present study and of this initial version of LUCATOO.

- **Specific comment 1 (L47): While forest loss is a prominent example, it might be worth highlighting that the impacts of LULCC are not just limited to forests but can affect a diverse range of ecosystems. For example: peatland drainage resulting in carbon emissions, soil carbon loss due to overgrazing of semi-natural grasslands. Additionally, LULCC impacts aren't necessarily binary (e.g. forest vs. no forest) but can exist on a continuum based on different levels of land use intensity. While this study focuses on absolute deforestation and afforestation, it's important to note the complexity of forest degradation more generally.**

  **Response:** We fully agree with the reviewer that the sole focus on forest ecosystems is not enough to highlight the diverse and severe impacts of LULCC. In this study, we focus exclusively on forest biomes, following the current definition of the planetary boundary for land system change (PB-LSC), which is the prime motivation for the development of our tool. Steffen et al. 2015 justify the focus on forest biomes in the following way: *"the control variable has been changed from the amount of cropland to the amount of forest cover remaining, as the three major forest biomes—tropical, temperate and boreal—play a stronger role in land surface–climate coupling than other biomes".* To address the reviewer's concern, we will revise the Introduction to clearly state that the definition of LULCC adopted in this paper is tailored to PB-LSC and therefore focuses exclusively on forest transformation. In addition, we will add a dedicated paragraph to the Discussion that explicitly acknowledges the resulting limitations. In this section, we will clarify that LUCATOO is tailored to the current PB-LSC definition and thus excludes changes in other ecologically and climatically important land systems such as savannas, grasslands, or wetlands, a conceptual limitation inherited from the PB-LSC framework rather than a modeling choice specific to LUCATOO. In our manuscript, we will note that future versions of the tool could be adapted to include additional land systems. Furthermore, we will include that by omitting more nuanced forms of forest degradation such as selective logging, fragmentation, or understory thinning

LUCATOO likely underestimates the full spectrum of anthropogenic pressures on forest ecosystems and their Earth system impacts.

- **Specific comment 2 (L50-54): A little more background about how the 50% and 85% boundaries were chosen would be useful. It's not clear how the third sentence of that paragraph leads on from the previous two. Where are we currently with respect to these boundaries? I think there needs to be more justification for setting a lower boundary for temperate forests or at least a discussion of the limitations of this assumption. The original source (Steffen et al. 2015) states "this is a provisional boundary only" – are there no recent updates? This lower boundary is justified in Steffen et al. by referencing Snyder et al 2004. However, I find this claim is not at all clear just from that paper - the climate impacts of removing temperate forests seems comparable to other biomes.**

  **Response:** The current definition of PB-LSC is still based on provisional estimates, a shortcoming that we had already acknowledged in the original manuscript (ln 50). We do agree with the reviewer that the rationale behind the placement of the 50% and 85% boundaries needs to be stated in the paper. In our revised manuscript, we will elaborate on the rationale behind the different boundary values assigned to the major forest biomes. Specifically, we will explain that the stricter precaution applied to tropical and boreal forests reflects their strong and distinct climate feedbacks (moisture provision, albedo feedback). This shortcoming builds one of the major incentives of the study and LUCATOOv1: to be able to selectively deforest and reforest and quantify the impacts on affected Earth system processes (Fig 1).

- **Specific comment 3 (L167-169): Why were continents chosen as the regional boundaries? That seems somewhat arbitrary. Wouldn't ecologically relevant boundaries such as ecoregions be more appropriate? Or national boundaries given the importance of domestic policies.**

  **Response:** We follow the original control variable definition from Steffen et al. (2015) who state: *"In particular, we focus on those land-system changes that can influence the climate in regions beyond the region where the land-system change occurred."* This line of argumentation builds the premise for analyzing LULCC on a continental

scale. We thank the reviewer for this important comment, as it links to one of the novelties and strengths of our approach: with the introduction of LUCATOO, the scientific community is given a tool at hand to adjust LULCC on a chosen spatial level. We have already explored this possibility and adjusted LULCC on the level of IPCC reference regions (see *Appendix C: IPCC reference region level*). Moreover, we conducted first test runs to operate LUCATOO on the national level but the spatial resolution of the underlying LPJmL model (0,5°x0,5°) constitutes a challenge for such finer granularity.

- **Specific comment 4 (L197): Uniform intensification is rather unrealistic. LULCC is influenced by a range of factors including land productivity and costs, national and international demand for commodities, and proximity to existing managed lands. This leads to complex LULCC patterns which are rarely spatially uniform, particularly on continental scales. A discussion on the limitations of this assumption is needed.**
  **Response:** Yes, as already stated above, we fully agree that the patterns of anthropogenic action driving LULCC are far more complex than the uniform intensification featured in LUCATOO. See above comment on the to be added limitation subsection in the discussion which we extend to include a paragraph on the limitations of performing a uniform intensification.

- **Specific comment 5 (L215): Figure 5. The notation used in the figure could be improved. It's not clear whether i, m, p etc. are parameters, variables or sets. For example, using the key in the top right corner, I would translate "i < LU_scn" as "subset of scenario dataset is less than scenario dataset" – while I can guess the intended meaning with the help of the caption, it's perhaps a bit unconventional. Maybe it would be clearer with something like Si < SLU_scn where S is the LSC boundary variable. "m = i*fac.re[m]" and similar is particularly difficult to parse – is this representing the transformation of subset i into subset m? In which case, perhaps this could be written as "m = fre(i)" where fre() is the reduction function?**
  **Response:** This is an important critique. In the revised manuscript, we will improve the notation in Fig. 5 to make the logic of the allocation procedure more transparent

and conventional. Specifically, we will revise the symbols to clearly distinguish between land-system state variables, subsets of land-cover fractions, and transformation steps, and we will avoid relational notation that could be misread.

- **Specific comment 6 (L226-230): Some background information on previous assessments would fit well in the introduction (see comment for L50-54).**
  **Response:** We will include and discuss the following references in the introduction: Richardson et al. (2023), more detailed and biome-specific results of their findings. Tobian et al. (2024), who show that temperate forests are more resilient to future climate change than boreal and tropical forest biomes in the LPJmL model. This study found boreal dieback under severe climate change and changes in the PFT distribution of the tropical forest biome (shift from evergreen to deciduous tree types).

- **Specific comment 7 (L255): Figure 6. Higher resolution image needed. It's interesting that each scenario shows either reforestation (planetary boundary) or deforestation (risk and strong transgression) in all biomes but not a mixture of both. Why is that? Given that different biomes in different continents are at or below the planetary boundaries (Table 1), shouldn't result in a more heterogenous response? Also, the uniform application of the intensity factors within each biome is very apparent here. I think there needs to be discussion whether this is realistic, given that observed LULCC is spatially (and temporally) heterogeneous.**
  **Response:** The reason for this uniform change is the 'prohibiting deforestation' flag, which (turned on for the datasets shown in Fig 6). Currently, the flag is described in line 191 and 239 of the manuscript. This flag protects cells from being subject to deforestation (that includes both the intensification and expansion operations) if their current value is above the boundary threshold. In other words, if the scenario is focused on forest restoration (as it is the case with scenario (I) - Planetary Boundary), deforestation will not occur if the regional biome value is above the PB-LSC threshold. We will highlight the flag in the caption of Fig 6. Regarding the uniform application, see our comment to "Specific comment 4 (L197)(RC1)".

- **Specific comment 8 (L265-267): It's not clear how this has been demonstrated. You have produced maps consistent with the PB-LSC boundary but there was no further analysis of how other PBs are affected under this scenario. Or is this referencing Richardson et al. 2023 (as it appears so further down)?**

  **Response:** It is correct that we have not demonstrated this. We fully agree with the reviewer that the current phrasing "can help to" is not strong enough to highlight that this paper can enable future research to stress-test the boundary value placements. This section will be largely rewritten to better align with the improved introduction.

- **Specific comment 9 (L289): What did Drüke et al. 2024 find?**

  **Response:** We will elaborate the important findings by Drüke et al. (2024) who thoroughly examined how the land system change scenarios derived from LUCATOO would affect the Earth system in the long run. They found that a further violation causes a considerable loss of carbon from vegetation and raises global temperatures and aridity.

- **Specific comment 10 (L295-298 and L306): This is an important point of discussion that should be expanded on (also see previous comment).**

  **Response:** We will expand the relevant parts of the manuscript to more explicitly address the applicability and limitations of the approach across spatial scales. In particular, we will add a clear reference to Section C of the Appendix, where the application of LUCATOO at the level of IPCC reference regions is demonstrated.For more detail refer to answer to Specific comment 3 (L167-169)(RC1).

- **Specific comment 11 (L310-321): As previously commented, the reallocation of CFTs based purely on area is an important limitation here. A more detailed allocation tool would consider other factors such as potential yields, land suitability and production costs as well as trade-offs between agricultural expansion and intensification. Similarly, afforestation could be prioritised based on preservation of ecosystem services such as biodiversity and carbon storage. On a more fundamental level, it's also not clear whether the scenarios presented here are internally consistent – for example, is the amount of deforestation in the strong transgression scenario even feasible given**

**socioeconomic constraints? How much demand growth (food, timber etc.) would be required to cause this much deforestation?**

**Response:** We thank the reviewer for this constructive comment. We fully agree that CFT reallocation purely based on area is a strong simplification (see the now added section on limitations as a response to your earlier comment). A future iteration of LUCATOO could incorporate socioeconomic constraints such as potential yields, production costs, and trade-offs between agricultural expansion and intensification. Prioritizing reforestation based on ecosystem service provision (e.g. biodiversity, carbon storage, or hydrological regulation) is an interesting option as this would allow for more realistic or policy-relevant land-use patterns and will subsequently be included in the discussion part. Regarding the internal consistency, please refer to our reply to General comment 1(RC1) "we will emphasize that the resulting scenarios should be interpreted as stylized, boundary-consistent land-system states rather than plausible future trajectories".

- **Specific comment 12 (L329): "cannot be adjusted or modified to depict specific anthropogenic pressure levels" – to the contrary, many land system models work explicitly with "anthropogenic pressure levels", although these can be expressed in different ways (e.g. demand for commodities, marginal utility of ecosystem services). Prominent examples include the major IAMs (IMAGE, REMIND-MAgPIE etc.) and other frameworks such as LandSyMM. While these haven't extensively explored the PB framework, there's no reason why PB-oriented scenarios couldn't be constructed within these models.**

  **Response:** This is an important clarification, and we agree with the reviewer that many IAMs explicitly represent anthropogenic pressures and, in principle, could be used to construct scenarios relevant to the PB framework. Our original statement was not intended to suggest that such models are incapable of representing anthropogenic pressure per se. Rather, following the PB-LSC definition, anthropogenic pressure is expressed specifically in terms of remaining biome-specific forest extent relative to PNV, enforced at regional scales. Systematically varying such constraints within IAMs typically requires substantial model-specific modifications, additional assumptions, and iterative tuning of socio-economic drivers.

To clarify this distinction and avoid misinterpretation, we will revise the manuscript text accordingly. In particular, we will replace the original sentence with the following formulation: "Currently available LULCC scenario products are generally not designed to directly align with the definition of PB-LSC, nor to systematically vary land-system pressure as defined by biome-specific remaining forest extent relative to potential natural vegetation at regional scales. While such configurations could in principle be constructed within existing land-system and integrated assessment models, doing so typically requires substantial model-specific adaptations, additional assumptions, and iterative tuning of socio-economic drivers."

We will further clarify that, while LUCATOO is limited in its representation of socio-economic complexity, it provides direct and transparent control over the land-system state required to study the Earth system impacts of varying transgression levels of PB-LSC.

- **Technical corrections 1: (L65) Replace "allocation models" with "land use models"**
  **Response:** Will be corrected accordingly.

- **Technical corrections 2: (L90) "The following _" section?**
  **Response:** Will be corrected accordingly.

- **Technical corrections 3: (L260-264) Too repetitive and non-specific, particularly "bridge the conceptual gap of an adjustable depiction"**
  **Response:** We will shorten the paragraph and make it more concise. It will read: *"Being easily extendable and reproducible, LUCATOO is a versatile tool that ensures the consistent and spatially explicit mapping of different PB-LSC statuses while maintaining the flexibility to be employed for various applications outside the PB framework context."*

---

## Author Comment (AC2)

Replies to **Reviewer 2:**

- **General comment 1: In their manuscript, entitled „LUCATOOv1 – A new land use change allocation tool and its application to the planetary boundary for land system change with the LPJmL model", the authors describe the LUCATOO model, a land use change allocation model which can directly be applied on the planetary boundary approach.**

  **This is an important research topic, since drivers of land-use change are often not well represented in existing approaches and their impacts and interlinkages on various environmental systems are complex and understudied. The direct link of this approach to the established framework of the planetary boundaries (PBs) potentially allows to better understand combined land use change impacts on PBs.**

  **The paper is well written and well structured; it also suits well to the journal "Geoscientific Model Development". Nevertheless, there are several major methodological aspects that in my opinion should be clarified.**

  **First, it should be described in more detail, where the BP-LSC values (85% tropical, 50% temperate, 85% boreal) for the scenario I (at the planetary boundary) exactly come from and how they were estimated or if this is just an arbitrary assumption. I also tried to find out in Richardson et al. 2023 and found the numbers there, but not an explanation how they were quantified. To me, these assumptions largely determine the results and therefore require a more detailed explanation.**

  **Response:** Yes, the current boundary definition is based on critical assumptions that strongly influence the results. The definition has not been changed by Richardson et al. 2023 and dates back to the last PB assessment by Steffen et al. 2015. Steffen et al. 2015 justify the boundary values in the following way: *"Of the forest biomes, tropical forests have substantial feedbacks to climate through changes in evapotranspiration when they are converted to nonforested systems, and changes in the distribution of boreal forests affect the albedo of the land surface and*

*hence regional energy exchange. Both have strong regional and global teleconnections. The biome-level boundary for these two types of forest have been set at 85% (Table 1 and the supplementary materials), and the boundary for temperate forests has been proposed at 50% of potential forest cover, because changes to tem- perate forests are estimated to have weaker influences on the climate system at the global level than changes to the other two major forest biomes."* They clearly state the provisional nature of these values. In their supplementary material, Steffen et al. (2015) further state that "This is also a provisional boundary, as there is no equivalent research on the boreal forest biome (as for tropical forests) exploring where thresholds might lie in terms of the fraction of forest converted before self-reinforcing feedback mechanisms are activated." The provisional nature of the threshold values is a large motivation for the development of our tool, as LUCATOO can largely help to scrutinize the underlying assumptions as it allows for the creation of land-use datasets that are tailored to the definition of PB-LSC. Our contribution is therefore not to defend these thresholds, but to provide a tool that enables the necessary research on systematic, spatially explicit testing of their Earth system implications which can lead to an improved boundary definition.

- **General comment 2: In the same way, it is not clear to me, how the values for scenario II (60% tropical, 30% temperate, 60% boral) and scenario III (40% tropical, 20% temperate, 40% boreal) were set. Are there any reasons, assumptions or other studies applying Earth System models underlying these values? Further, given the high uncertainty to these numbers, the authors should apply a sensitivity analysis to demonstrate the sensitivity of the model for different assumptions.**

  **Response:** We thank the reviewer for the opportunity to clarify these important aspects. The values for scenario II (60% tropical, 30% temperate, 60% boreal) are the provisional values set by Steffen et al. 2015 for the infliction point towards an Earth system state that poses significantly higher risk gradients. The values for Scenario III are those when LUCATOO was applied in Richardson et al. 2023. Fig. 2 of that study shows the detrimental effects of this scenario on global temperature development and terrestrial carbon storage. We further fully agree with the reviewers remark on the sensitivity analysis but would like to highlight that such an analysis

should be conducted with an Earth system model such as the Potsdam Earth Model (POEM, Drüke et al., 2021), to better understand the impacts on the Earth system that are inflicted by changes in the transgression level of PB-LSC. We will encourage such an application in the discussion part of the manuscript.

- **General comment 3: Another question in this context: Why do temperate forests have a much lower value for the 'safe operating space'? Is this because they have already been deforested by a large extent?**
  **Response:** The justification by Steffen et al. 2015 (as can be found in their Supplement) states: *"Both tropical forests (changing evapotranspiration) and boreal forests (changing albedo) have strong impacts on the climate system with global teleconnections from the regional changes, while temperate forests are assessed to have only moderate influence on the global climate"*. Again, our methodical contribution is not to re-derive or fortify these thresholds but to create a tool that can create datasets to stress-test the underlying assumptions.

- **General comment 4: Another major issue affects the definition of the intensification scenario, that in my opinion is also a specific form of expansion, but within regions that already have cultivated areas. Thus, the spatial scale of the approach determines if agricultural expansion is interpreted as intensification or expansion, which can be largely misleading. The main methodological issue is that the authors uniformly apply intensification or afforestation factors across biomes. This is very unrealistic. Also compared to other land use models, I don't see any theory behind this allocation algorithm. Other land use models (e.g. Globiom) consider a large range of different factors that spatially vary widely, such as capital, labour, or land productivity (in Globiom, this is also provided by a mechanistic crop model), and costs (e.g. fertilizer costs for intensification that have very different costs in different regions). Most land use models are coupled with CGE or PE models to consider dynamically changing patterns of supply and demand for different regions but also global trade between these regions. All this is not considered in the LUCATOO but would largely impact on the spatial patterns and the degree of both, intensification and expansion.**

**Therefore, I would be careful applying the model for impact assessments of LULCC on PBs. The identified spatial patterns matter a lot. Therefore, approaches that investigate possible impacts on the environment or on PBs or analyse other trade-offs (e.g. with biodiversity, food security, carbon sequestration, GHG emissions, etc.) should be able to consider main drivers of land use change.**

**That said, the major weakness of this study is a lack of the validation of the approach that demonstrate e.g. if simulated land-use change patterns for historical periods are realistic and can be reproduced.**

**Response:** We thank the reviewer for this detailed and important critique and largely agree with the points raised. In particular, we acknowledge that the distinction between agricultural expansion and intensification depends on spatial scale, and that the intensification scenario implemented here can indeed be interpreted as a form of expansion within already cultivated regions. Our use of the term "intensification" therefore refers to increasing agricultural area within regions that are already managed, rather than yield increases through higher inputs. We will clarify this definition more explicitly in the revised manuscript to avoid ambiguity.

We also fully agree that the uniform application of intensification or afforestation factors across biomes is a strong simplification. As the reviewer correctly notes, spatial patterns of land-use change in reality are shaped by a wide range of regionally heterogeneous socio-economic factors that are explicitly represented in land-use and integrated assessment models such as GLOBIOM, IMAGE, or REMIND–MAgPIE. The allocation algorithm of LUCATOO is intentionally simple and transparent, precisely because the tool is designed as a diagnosis tool for controlled Earth system experiments rather than for realistic simulation of land-use decision-making. By abstracting from socio-economic drivers, LUCATOO allows the spatial extent of land-system change to be varied in a systematic and reproducible way, tailored to the PB-LSC definition and without introducing additional, model-specific economic assumptions. We will include a limitation section highlighting the lack of representation of underlying socio-economic drivers in this current version of LUCATOO.

- **General comment 5: Another important point that should be considered by the authors and added to the discussion is that land use requirements are only defined by conservation goals in this study. Other sustainability goals, such as food security (SDG2) or renewable energy production are not captured by the PBs? The results of this analysis therefore could lead to trade-off with different SDGs that not considered within the PB approach.**

  **Response:** Yes, other sustainability goals as those mentioned by the reviewer are not explicitly represented and trade-offs between conservation goals and other SDGs are likely but not addressed within the scope of this study. In this study, we stringently follow a central premise of the planetary boundaries framework: to identify biophysical limits that should not be exceeded in order to maintain Earth system stability. In consequence, LUCATOO inherits this focus and allows for the creation of land-use data sets that are consistent with specified PB-LSC levels, without evaluating whether these states are compatible with food production requirements, bioenergy demand, or other development goals. A further alignment of these sustainability objectives, i.e. how planetary boundary interactions affect various SDGs, is an underexplored realm of research.

  To make this restriction clear, we will include the following in the discussion: *"In this study, the current biome-level definition of PB-LSC is the only factor used to define LULCC requirements. Therefore only conservation-oriented goals pertaining to the stability of the Earth system are reflected. Potential trade-offs with other sustainability objectives, like the production of renewable energy or food security, are thus not explicitly considered."*

- **Specific comment 1: Ln 16: To state that the model 'can accurately represent the spatial distribution of agricultural land use for different statuses …', a model validation would be required that shows that the model is really able to reproduce historical land use transitions. This is however not done by the authors. Due to the lack of a validation approach, I also wonder why the authors use the term 'accurately' in this context.**

  **Response:** The term 'accurately' is being used in a strictly quantitative fashion in relation to the PB-LSC definition; i.e., the purpose of the tool is to accurately produce land-use datasets that result in changes of the remainder of forest biome

extent. We agree that this phrasing can be misunderstood and have now changed the word to 'reliably'.

- **Comment 2: Ln 20: instead of saying 'is openly accessible', I recommend providing the exact license here (e.g. CC BY 4.0).**
  **Response:** Will be added accordingly.

- **Comment 3: Ln 62f: To assess the impact on the stability of the Earth system, the land-use model would need to be coupled with a global Earth System model. This could also help to quantify the unknown extent to which forest cover must remain intact to sustain safe planetary conditions (as mentioned in line 50). As I understand, this is then again referred to in line 83-86. Maybe, this could be kept together?**
  **Response:** We will elaborate on the premises behind the PB-LSC definition in the introduction by adding: *"the exact placement of the boundary for each biome is still uncertain as it has not been systematically stress-tested across a wide range of spatially explicit land-use configurations"*. We do not agree that ln 62f and ln 83-86 should be merged. Ln 62f explains why LULCC scenarios are needed, while line 83-86 displays how our tool can actually do it, as shown in Richardson et al. 2023. We will change *"analysis of the consequences"* to *"improved understanding of the consequences"* in ln 62 to further improve clarity. We further strive for the tool to be consistently coupled with other new tools developed around LPJmL, i.e. the boundaries software package (Gerten et al. 2025; http://dx.doi.org/10.1016/j.oneear.2025.101341)

- **Comment 4: Ln 71: There are approaches that investigate impacts of agricultural intensification various key agricultural externalities, e.g. Folberth et al. 2020 (doi: https://doi.org/10.1038/s41893-020-0505-x), and approaches that assess trade-offs for agricultural expansion, e.g. Schneider et al. 2025 (doi: https://doi.org/10.1038/s41893-024-01410-x), and approaches that look at trade-offs between agricultural expansion, intensification, biodiversity and GHG emissions, e.g. Zabel et al. 2019 (doi: https://doi.org/10.1038/s41467-019-10775-z).**

**Response:** We thank the reviewer for pointing us towards these relevant articles and will include them in the introduction part (ln 71f) of our revised manuscript, by stating: *"The necessity to examine future modifications to the land system across a range of spatial and temporal scales and biomes is further underscored by expected increases in global cropland expansion to meet future food demand (Folberth et al. 2020) and the arising trade-offs with biodiversity protection targets and climate change mitigation goals (Schneider et al. 2024; Zabel et al. 2019)."*

- **Comment 5: Ln 79: Could you cite an article here that demonstrates that LUCATOO has the 'potential to be used for …'.**

  **Response:** As of now, LUCATOO has not been employed outside the context of the PB framework. We have, however, detailed how it *has the potential to be used for generating LULCC patterns for other scientific inquiries* in the discussion section of the paper, e.g. to study the effects of land-use change on precipitationsheds.

- **Comment 6: Ln 93: 'match' here means that it does not exceed the PB-LSC?**

  **Response:** Yes. The created dataset is corresponding with the values taken from the current PB-LSC definition, i.e. the forest remainder for each biome on each continent is matching this value.

- **Comment 7: Ln 90-102: Where do these numbers come from. Are these realistic scenarios? Until when – there is no time period determined. Particularly for scenario III, I wonder if there is any socio-economic basis for this assumption?**

  **Response:** We agree that the numerical values defining scenarios I–III, and particularly the strong transgression scenario (scenario III), require clearer explanation regarding their origin, interpretation, and intended use. The scenario values are not derived from socio-economic projections, nor are they associated with a specific time horizon. Instead, they are defined relative to PB-LSC and represent discrete land-system states characterized by different degrees of boundary transgression. Accordingly, these scenarios should not be interpreted as realistic land-use futures which would be constrained by socio-economic processes such as demand growth, technological change, or trade dynamics. The scenario

values are aligned with the current PB definition and have been already discussed in our response to the reviewers General comment 1.

For improved clarification, we will add the following section to the manuscript: *"The scenarios defined here represent static states of the land system characterized by varying degrees of PB-LSC transgression and are not linked to a specific time horizon. They are not intended to represent socioeconomically plausible future scenarios of land use, but rather to serve as stylized configurations for experiments on the sensitivity of the Earth system. In particular, the strong transgression scenario represents an extreme stress test and should not be interpreted as a probable or feasible outcome of future land use dynamics."*

- **Comment 8: Figure 2: It should also be added to the figure caption of Figure 2, that the values refer to the global biome areas. As such, 'safe operating space' also refers to impacts on the global Earth System and does not mean that there could be severe regional risks. Maybe this should be added to the discussion for clarification.**

  **Response:** We want to clarify that the land-system change boundary definition applied in the context of our study (as inherited by Steffen et al. 2015) focuses on biome area on a regional scale. This becomes clear when looking at the numbers provided in Table 1 by Richardson et al. (2023) or as shown in our Table 1 where we found that, for example, the boundary value for temperate forest in Europe is currently further transgressed than in Oceania. The rationale is that the loss of major forest biomes on any of the continents of their occurrence has implications for the stability of the Earth system as a whole (e.g. due to climate feedbacks, moisture provision etc.).

- **Comment 9: Ln 125f: Several questions on the LPJmL setup: What spatial dataset do you use for different managements, such as irrigation, fertilizer application, sowing dates for the agricultural crops? Is management kept constant over time or does it change over time? How is this handled in scenarios (that are not simulated in this study). Do land-use changes depend on management? This is not getting clear in the entire manuscript.**

**Response:** Being a DGVM, LPJmL requires forcing from exogenous datasets. Land-use, management and irrigation are obtained from Frieler et al. (2017) who provide spatially explicit crop management data on an annual basis. All external datasets will be referenced in the updated manuscript. The development of cropland over the historical period changes over time until the year 2015, in accordance with HYDE 3.2. (Goldewijk et al. 2017). Regarding the sowing dates, Schaphoff et al. 2024 (http://dx.doi.org/10.5194/gmd-11-1343-2018) state that these "are simulated based on a set of rules depending on climate- and crop-specific thresholds as described in Waha et al. (2012)." Land-use change, in terms of CFT occurrence per cell in a certain year over the historical period, is defined by the land-use data set mentioned above.

- **Comment 10: Ln 128: Why is historical climate model data (GFDL-ESM4) used here and not reanalysis data, such as 'GSWP3-W5E5'? Which time period is simulated?**

  **Response:** The historical time period simulation ensues the spin-up protocol and starts in 1850 and ends in the year 2015, which builds the starting point for the land-use reallocation performed with LUCATOO (cf. Fig. 3 in the manuscript). The GFDL-ESM4 forcing was chosen since its transient climate response is close to the multimodel mean of the CMIP6 GCM ensemble, see Meehl et al. (2020) (http://dx.doi.org/10.1126/sciadv.aba1981). We chose GFDL-ESM4 to ensure consistency between the land-use scenarios and the broader Earth system modeling framework in which the experiments are embedded. LUCATOO could be used to generate the status quo of PB-LSC under current land use (i.e. at the end of the historical period, here 2015) for various GCMs of the CMIP6 protocol, enabling a sensitivity analysis of Earth system responses to historical CMIP6 climate forcing and associated land–climate interactions under boundary-consistent land-system states.

- **Comment 11: Figure 3: According to Fig. 3, LPJmL must run not only historical periods, but for the scenarios also future periods? This is not getting clear from the previous description of LPJmL simulations, in which the authors say that historical climate model data is used for the simulations. This is a bit**

**confusing, because no future scenarios are simulated in this study. Nevertheless, the scenarios refer to the future -> compare Fig. 3.**

**Response:** In line with our answer to your previous comment, the end of the historical simulation (here the year 2015) builds the starting point for the reallocation. The produced dataset are per se timeless, but should be interpolated over time when forcing a DGVM or ESM to compute their response to time-varying LULCC input (refer to the transient period in Drüke et al. 2024). We understand however, that Fig 3 can be misunderstood in this context and will improve its readability by highlighting that future simulations of land-system change are not subject of LUCATOO per se, but that LUCATOO enables future simulations to be conducted in other DGVM/ESM studies.

- **Comment 12: Ln 145-151: Not clear: How are crops allocated to cells (which mechanism is behind) and what management is assumed here?**
  **Response:** In LPJmL, historical patterns of crop allocation patterns and their irrigation management (i.e. the CFT distribution in cells) are prescribed by exogenous datasets. As referenced in the manuscript, historical land use data is obtained from Frieler et al. (2017), based on HYDE 3.2. (Goldewijk et al. 2017).

- **Comment 13: Ln 153: Why the first 30 years of the simulation period (1850 – 1879) and not using PI control forcings until 2100 as a reference, which would be more consistent?**
  **Response:** The PNV simulation is forced by historical climate data but without LULCC. The period was chosen due its pre-industrial climate to avoid the observed impact of climate change on land-systems (see for example Parmesan & Yohe (2003), http://dx.doi.org/10.1038/nature01286). This practice is in line with earlier studies (see for example Tobian et al. 2024 (https://doi.org/10.1088/1748-9326/ad40c2) and references therein).

  We did not use output from PI control as this run is conducted to spin-up the LPJmL model and to bring it into a semi-equilibrium state, prior to all simulations. We further cannot follow the reviewer on the suggestion of using control forcings of the year 2100 as a PNV reference period and wonder on which basis this remark is resting?

Even modest climate forcings leave a significant impact on the distribution of biome locations (see Fig 3 in Tobian et al. (2024), something we deliberately chose to avoid by using pre-industrial climate forcings in the first place.

- **Comment 14: Ln 158: This information comes late and fits better to section 2.1.**
  **Response:** We fully agree and will adjust it in the revised manuscript.

- **Comment 15: Ln 164-213: How does the model decide on which pixels intensification or expansion occurs? What mechanisms and theories are behind it? This is not explained.**
  **Response:** LUCATOO does not feature economic optimization, agent-based decision-making, or mechanistic land-use theory but allocates land-use based on a set of explicit, rule-based spatial heuristics designed to generate reproducible land-use configurations that satisfy predefined boundary-related area constraints. These heuristics are either "intensification" (i.e. the replacement of PFTs by CFTs in cells where CFTs already occur) or "expansion" (i.e. the replacement of PFTs by CFTs which were priorly not put under management). These rule-based, spatial operations are depicted in Fig 4. For improved clarity, we will elaborate on this in the limitation section of the discussion in the revised manuscript.

- **Comment 16: Ln 171: Is afforestation the right wording here (usually afforestation is done actively by humans) or is it reforestation, or forest restauration? Further, it should be added to the discussion for the afforestation case, that afforested areas require time to take on functions of primary ecosystems, which should be relevant for PB assessments. Are there any elasticity functions in LUCATOO that constrain the transformation between different land uses and covers over time?**
  **Response:** This is a well-founded terminological critique highlighting that the term afforestation is potentially misleading in the context of our scenarios. In LUCATOO, the conversion to forest is not driven by active management intervention, but a prescribed change in land cover (here, relative to potential natural vegetation extent). In this sense, reforestation or forest restoration would indeed be more appropriate

terminology. We have revised the manuscript accordingly and now use the term reforestation.

We also fully agree that restored or newly established forests require substantial time to recover the structure and functions of primary ecosystems. Our tool operates on land-system states rather than trajectories. The important temporal dynamics (e.g. delays in ecosystem recovery) need to be taken into account in application studies that employ LUCATOO and create transient runs covering long time horizons. As has been done by Drüke et al. (2024).

We will expand the discussion on the limitations to emphasize this point. The planned addition will read: *"The restored forest areas in our scenarios represent potential forest cover rather than mature primary forest. Studies employing datasets created by the tool should feature long-term simulation runs to account for these temporal dynamics as has been done by Drüke et al. (2024)."*

- **Comment 17: Ln 172 and Ln 196: 'Deforestation can result from intensifying LULCC in cells where CFTs are already present, or by expanding LULCC into currently pristine cells (where CFTs are absent).' and 'Here, LULCC intensification replaces PFTs with CFTs'.**

  **I understand that intensification often comes with a compaction of land and more monocultures. Nevertheless, the conversion of forest into cropland is an expansion by definition. Anything else is just an effect of the spatial scale of the applied approach (approx. 50 km2 in this study). To me, both cases are an expansion of cropland and do not refer to intensification.**

  **Response:** We thank the reviewer for this important remark and agree that the conversion of forest (or any natural vegetation) into cropland constitutes agricultural expansion, regardless of whether it occurs in grid cells that already contain cropland. We will add a section in the method section of the manuscript to avoid confusion and improve clarity: *"Deforestation occurs either through cropland intensification within grid cells that already contain CFTs or through expansion into previously unmanaged grid cells where CFTs are initially absent. In both cases (intensification and expansion) natural vegetation (PFTs) is replaced by cropland (CFTs), and the distinction reflects the spatial scale of the allocation rather than different land-use processes."* From a methodological perspective, both terms are

required as the algorithm operates very differently if CFTs are already present in a cell or not (see Fig 4).

- **Comment 18: Ln 190-200: Intensification and afforestation are applied uniformly across all cells of the biome. Given the spatial heterogeneity of both soils and climate, but also the different socio-economies, the potential for afforestation and intensification varies strongly across regions within a biome. Why not using yields from LPJmL under different degrees of intensification levels (e.g using fertilizer application) or potential yields that can be used as a maximum threshold for intensification and to calculate yield gaps that can be closed by a certain amount for different intensification scenarios. This would result in much more realistic spatial patterns of intensification. Not clear if other feedbacks of intensification on PBs are considered, e.g. due to higher use of pesticides and fertilizers?**
  **Response:** We fully agree that the potential for both cropland expansion reforestation varies strongly within biomes due to their heterogeneity in climate, soils, productivity, and socio-economic conditions. We welcome the idea of constraining intensification through yield potential can be done in a future version of the tool (will be added to the limitation and outlook session of the manuscript).
  Higher levels of agricultural intensification will likely introduce additional pressures relevant to other PBs, such as nutrient loading or freshwater use. In the terminology of the PB framework, these constitute PB interactions. This is an important remark and will add a discussion point on how assumptions behind the applied intensification shape the spatial patterns and potentially the strength of PB interactions.

- **Comment 19: Ln 205: How is bioclimatic suitability of the CFT assessed? Please cite data/paper. Why is this done? Is this done at subscale within a half degree grid cell? Are just wetlands and waterways excluded, or also water bodies such as lakes and rivers? What about impervious surfaces, cities, etc.?**
  **Response:** The bioclimatic suitability is based on the CFT parametrisation of LPJmL (see for example chapter 2.4 in Schaphoff et al. (2018) - LPJmL4 – a dynamic global vegetation model with managed land – Part 1: Model description) on a cell-based

level: it is checked whether an entire cell fulfils the bioclimatic suitability conditions required by the CFT-specific parametrisation. The reason why we constrained LULCC by bioclimatic suitability is to avoid CFT allocation into areas which are strictly not suitable for crop production. The same applies for wetlands which also cover lakes and rivers. The permiteability of a cell is based on soil properties which are static and prescribed. A reference to the dataset will be made in the updated manuscript.

- **Comment 20: Ln 210: Usually land use transitions are not linear over time. Often, sigmoidal functions describe this better.**
  **Response:** We thank the reviewer for this insightful comment. If required, we can adjust the code and include the option of a sigmoidal function for conducting the land use transition.

- **Comment 21: Ln 251: Suggest to refer to Fig. 6.**
  **Response:** Will be added.

- **Comment 22: Figure 6: Very small and difficult to see details in the maps. Also the resolution in the compiled pdf is insufficient to zoom in.**
  **Response:** We fully agree and decided to move the PFT maps to the appendix which gives more space in the figure to increase the size of the remaining map and ensure a high resolution of the submitted PDF.

- **Comment 23: Ln 265-267: Not clear, this study doesn't investigate the impacts on other PBs. Also, I don't agree that this study helps to corroborate the assumed thresholds.**
  **Response:** The initial statement was not clearly formulated and could have been slightly misleading. We have now deleted this sentence in the updated manuscript.

- **Comment 24: Ln 291-294: 'LUCATOO opens a new realm of research…'. I think that's a bit far-fetched. Many other land use models exist, that include a representation of land use change drivers – which LUTATOO is obviously**

**missing completely. Other existing land use models in principle can also be applied on PB impacts.**

**Response:** In the light of the discussion and prior remarks of the reviewer we agree and adjust the paragraph. It will be adjusted to: *"LUCATOO facilitates the required research on the impacts of LULCC in the context of planetary boundary interactions and systemic environmental impacts across various sectors of human activity."* Regarding the socio-economic drivers, we want to draw the reviewers attention again to the newly added limitation section.

- **Comment 25: Ln 295: 'Our land use change reallocation tool is easily extendable'. This has already been stated several times. To investigate other Earth System processes, I would not suggest using this model, due to the already mentioned weaknesses.**

  **Response:** With this sentence we wanted to highlight that the simplicity and modality of our approach can allow for applications in Earth system research where more complex land-use models are less easy to be controlled. We agree that the statement in its current form is too strong and will adjust it accordingly: *"Our land use change reallocation tool is easily extendable and can, when taking its limitations into account, be extended to fit other applications that require stylized land-use configurations for controlled Earth system sensitivity experiments."* Moreover, we will ensure the readability of the manuscript by excluding reputative parts, e.g. ln 260: "Being easily extendable and reproducible, LUCATOO is a versatile tool".

- **Comment 26: Ln 300: Intact forest biomes are not necessarily hotspots of forest biodiversity. I would say 'and thus can be hotspots of …'.**
  **Response:** Will be corrected as suggested.

- **Comment 27: Ln 310: Similar assumptions are used by most of the existing land use models. Often, existing road networks are used to determine future agricultural expansion. Nevertheless, this assumption could quickly become obsolete, for example if a new road is built through the rainforest.**
  **Response:** We fully agree. In the context of this paper, the assumption is that our tool does not aim to predict future infrastructure development or dynamically

evolving accessibility patterns; a limitation that is consistent with the PB-LSC status-based, non-predictive design of the framework.

- **Comment 28: Ln 328: This reads a bit ignorant. Why can other approaches not be adjusted or modified to depict specific anthropogenic pressure levels? Actually, they already do.**

  **Response:** We will adjust this section and plan on rewriting it as follows: *"Currently available LULCC scenario products are generally not designed to directly align with the definition of PB-LSC, nor to systematically vary land-system pressure as defined by biome-specific remaining forest extent relative to potential natural vegetation at regional scales. While such configurations could in principle be constructed within existing land-system and integrated assessment models, doing so typically requires substantial model-specific adaptations, additional assumptions, and iterative tuning of socio-economic drivers."*